# Unusual Presentation of Feline Leprosy Caused by *Mycobacterium lepraemurium* in the Alpine Region

**DOI:** 10.3390/pathogens10060687

**Published:** 2021-06-01

**Authors:** Giovanni Ghielmetti, Sarah Schmitt, Ute Friedel, Franco Guscetti, Ladina Walser-Reinhardt

**Affiliations:** 1Institute for Food Safety and Hygiene, Section of Veterinary Bacteriology, Vetsuisse Faculty, University of Zurich, 8057 Zurich, Switzerland; sarah.schmitt@vetbakt.uzh.ch (S.S.); ute.friedel@vetbakt.uzh.ch (U.F.); 2Institute of Veterinary Pathology, Vetsuisse Faculty, University of Zurich, 8057 Zurich, Switzerland; gufo@vetpath.uzh.ch; 3Vetaugenblick.ch, Wiesentalstrasse 142, 7000 Chur, Switzerland; ladinawalser@gmail.com

**Keywords:** *Mycobacterium lepraemurium*, feline leprosy, cat, nontuberculous mycobacteria, cutaneous mycobacteriosis

## Abstract

A 9-year-old cat was referred with multiple, raised, ulcerative and non-ulcerative nodules in the periocular area, sclera and ear-base region, and on the ventral aspect of the tongue. In addition, a progressive ulcerative skin nodule on the tail was observed. Fine-needle aspirations of multiple nodules from the eyelid and sclera revealed the presence of histiocytes with numerous acid-fast intracellular bacilli. The replication of slowly growing mycobacteria in liquid media was detected from biopsied nodules after three months of incubation. The molecular characterization of the isolate identified *Mycobacterium (M.) lepraemurium* as the cause of the infection. The cat was treated with a combination of surgical excision and a four-week course of antimicrobial therapy including rifampicin combined with clarithromycin. This is an unusual manifestation of feline leprosy and the first molecularly confirmed *M. lepraemurium* infection in a cat with ocular involvement in Europe. The successful combination of a surgical and antimycobacterial treatment regimen is reported.

## 1. Introduction

The clinical significance of mycobacterial infections is increasing in human and veterinary medicine [1,2,3]. Mycobacteria-associated skin lesions in cats are currently classified into three groups according to different specific etiologies and include feline leprosy historically attributed to *Mycobacterium (M.) lepraemurium* infection, cutaneous tuberculosis caused by *M. tuberculosis* complex (MTBC) members and lesions due to facultative pathogenic nontuberculous mycobacteria (NTM) [4]. The line between these forms is, however, very subtle, from both the clinical and etiological points of view. By definition, feline-leprosy-causing NTM are fastidious microorganisms that are not necessarily phylogenetically related and form a heterogeneous group. Feline leprosy has been reported in the western part of the United States [5] and Canada [6,7], coastal Australia and New Zealand [8,9,10,11] and different European countries including the United Kingdom [12], the Netherlands [13], France [14], Greece [15] and Italy [16]. However, unequivocal identification of the species involved has not always been possible, and the diagnosis has often been based on clinical and histopathological findings, similarities with experimental models or after the detection of DNA belonging to the genus *Mycobacterium*. 

It was only after the advent of molecular methods that different mycobacterial species including *M. lepraemurium*, *M. visibile*, *Candidatus* ‘Mycobacterium tarwinense’ and *Candidatus* ‘Mycobacterium lepraefelis’ were identified as the etiological agents of feline leprosy [9,10,11,17]. Current reports show marked differences in the geographical distributions of the involved pathogens, with *Candidatus* ‘Mycobacterium tarwinense’ and *Candidatus* ‘Mycobacterium lepraefelis’ reported exclusively in Australia and New Zealand, where they cause most of the clinical infections, and *M. lepraemurium* being described worldwide in temperate maritime climates [4]. Recent genomic investigations showed that *M. lepraemurium* belongs to the *Mycobacterium avium* complex (MAC) [18], whereas Candidatus ‘Mycobacterium lepraefelis’ is genetically similar to the human pathogens *Mycobacterium leprae* and *Mycobacterium lepromatosis* [9], and Candidatus ‘Mycobacterium tarwinense’ is related to the *Mycobacterium simiae* group [11]. *Mycobacterium lepraemurium* has been described to be the causative agent of murine leprosy, and cats are believed to be infected through hunting diseased wild rodents [19].

Feline leprosy is characterized by the formation of single or multiple granulomatous skin nodules and is caused by mycobacteria species that cannot be cultured using routine methods. Clinically, feline leprosy is indistinguishable from cutaneous tuberculosis or other mycobacterioses. Due to the specific zoonotic potential and intrinsic pattern of resistance to antimicrobials found in some species, it is crucial to identify the causative agent involved whenever possible. Cat-to-human transmission has been reported in cases of *M. bovis* infection [20]; moreover, *M. microti* and some NTM, including *M. avium*-complex (MAC), *M. abscessus, M. nebraskense*, and *M. kansasii*, are potentially zoonotic [21,22,23]. Hereby, a case of feline leprosy originating from the Swiss Alpine region is presented. This is an unusual manifestation of *M. lepraemurium* infection and the first documented case with ocular involvement in Europe. The clinical and cytological findings, diagnosis, and treatment of this mycobacterial infection in a Maine Coon cat are described.

## 2. Case Description

A 9-year-old, 4.2 kg, female, neutered Maine Coon cat presenting with nodular skin lesions ranging in size from 5 to 20 mm around both eyes (Figure 1A,B), on the right ear-base region and on the ventral aspect of the tongue (Figure 1C,D), was referred to a small animal ophthalmologist. In addition, a progressive ulcerative skin nodule was observed on the tail (Figure 1E). The owner described the skin nodules as indolent and reported that the periocular and tail lesions had developed several weeks prior to presentation. 

The cat was born in Switzerland and was never moved to other countries. The Maine Coon lives in a Swiss Alpine Dfc climate region according to the Köppen–Geiger classification system, at an altitude of approximatively 1000 m above sea level. The average temperature and the monthly rainfall in the region are approximatively 1 °C and 70 L/m^2^ and 20 °C and 110 L/m^2^ in the winter and summer, respectively (www.meteoschweiz.admin.ch, accessed on 27 May 2021). No other cats share the same household; however, the Maine Coon has access to the outdoors and is a known hunter.

Ophthalmic examination revealed an ulcerative nodular skin lesion on the left lower eyelid and non-ulcerative nodules on the left dorsolateral sclera and the right medial canthus (Figure 1A,B). Dazzle and palpebral reflexes were intact bilaterally. The menace responses were positive on both eyes. The results for the Schirmer tear test and intraocular pressure were within normal limits, as was the fluorescein dye test result, which was negative in both eyes. There was no history of any prior ophthalmic disease, or immunosuppressive or immunomodulatory medications, and no evidence of current or previous intraocular inflammation. The cat had no significant hematological or biochemical alterations, and blood tests for feline leukemia virus and feline immunodeficiency virus were negative (SNAP FIV/FeLV Combo Test, Idexx). The animal was alert and responsive during physical examination, with a body condition score of 7/9 and vital parameters within the reference limits. Abdominal ultrasound and a thoracic latero-lateral survey radiograph showed no abnormalities.

Differential diagnoses for palpebral nodules include neoplasia such as squamous cell carcinoma, mast cell tumors, hemangiosarcoma, adenocarcinoma, peripheral nerve sheath tumors, lymphoma, apocrine hidrocystoma, or hemangioma [24]; infection with bacterial, mycotic or parasitic agents including *Cryptococcus* species and *Leishmania infantum* [25]; or single and multiple cysts [26,27].

## 3. Results

Cytopathological fine-needle aspirates of the lesions on the left lower eyelid and the sclera stained with an automated stainer revealed a mixed inflammatory cell infiltrate consisting of neutrophils and of a large number of macrophages containing numerous intracytoplasmic rod-shaped, partly darkly blue-stained and partly unstained bacteria that were acid-fast in the Ziehl–Neelsen stain (Figure 2A,B). A cutaneous mycobacteriosis was suspected. Thus, tongue, tail, and ear-base lesions were not further investigated.

Based on the cytological findings, the palpebral nodule on the left lower eyelid and the scleral lesion were surgically removed through a wedge and marginal resection, respectively. The wedge resection was closed with an absorbable Vicryl 6-0 suture, whereas the scleral resection was left open. Afterwards, cryotherapy using carbon dioxide was performed on the scleral side. Initially, no treatment for the nodular lesions on the ventral aspect of the tongue, tail and ear base were undertaken. The resected nodules were subjected to mycobacterial culture in sterile saline and processed as previously described [28,29]. Briefly, the samples were homogenized and decontaminated using H_2_SO_4_ (4%), followed by neutralization with NaOH (1M). Two BBL MGIT liquid medium tubes supplemented with Bactec MGIT 960 growth supplement, BBL MGIT PANTA (polymyxin B, amphotericin B, nalidixic acid, trimethoprim and azlocillin) antibiotic mixture (Becton, Dickinson, BD) and 50 μg/mL of sodium pyruvate were inoculated with 0.5 mL of homogenized and decontaminated specimen. In addition, two Löwenstein–Jensen and Stonebrink agar slants (BD) were inoculated with the same inoculum and incubated for up to three months. One set of culture media was incubated at 37 °C, and the other was set at 30 °C.

Growth of mycobacteria was detected in liquid medium (BBL MGIT) at 37 °C after approximately 90 days of incubation. No growth could be observed on solid media. Visible colonies were seen on subcultures from liquid media on selective egg-yolk medium with pyruvate (Artelt-Enclit) after 30 days of incubation at 37 °C. The amplification and sequencing of the 16S rRNA and *hsp65* genes revealed the presence of *M. lepraemurium* species DNA with identity scores of 533/534 bp and 1530/1530, respectively, compared to *M. lepraemurium* strain Hawaii available in NCBI under accession number CP021238 [30].

Once the diagnosis of granulomatous dermatitis and scleritis caused by *M. lepraemurium* was confirmed through molecular findings approximately 120 days after surgery, the patient was treated with a four-week course of antimicrobial therapy including rifampicin (Rimactan, Sandoz Pharmaceutical) at 75 mg (approx. 18 mg/kg) q24h combined with clarithromycin (Clarithromycin-Mepha) at 50 mg (approx. 12 mg/kg) q12h. Due to the extremely slow growth of the bacteria in vitro, no antimicrobial susceptibility testing was attempted. Since the cat was not cooperative and the lesions did not recur, the owners independently decided to stop the antimicrobial therapy after the completion of the initial four weeks. During this treatment period, no side effects were observed. Whilst the lesion on the tail was surgically removed, interestingly, the nodular lesions on the ventral aspect of the tongue healed spontaneously before antimicrobial therapy was started (Figure 3A,B). The ear-base lesion was still present when antimicrobial therapy was started. One year after the surgical resection of the lesions and six months after the discontinuation of the antimicrobials, the cat was mentally alert with a good appetite. By clinical examination, the cat’s ocular and dermatologic lesions, including the ear-base lesion, were considered resolved. All the wounds healed with almost no scar formation, and no nodular lesions had reappeared at the time of writing (Figure 3A,B).

## 4. Discussion

To date, more than 180 species of NTM have been isolated worldwide, and more than 60% of these are known to be pathogenic to humans or animals [31,32]. Among the NTM, members of the MAC are the leading cause of pulmonary disease and disseminated mycobacteriosis in immunocompromised human patients. *Mycobacterium avium* subspecies *paratuberculosis* (MAP), a member of the MAC, is the etiological agent of paratuberculosis, a contagious disease listed by the World Organisation for Animal Health. This chronic progressive intestinal disease primarily affects ruminant species and has also been reported in horses, pigs, alpacas, llamas, rabbits, free-ranging carnivores and dogs [33,34,35]. 

*M. lepraemurium* is an extremely slow-growing, acid-fast bacillus first reported in brown rats [36]. Recent genomic investigations classified *M. lepraemurium* as a member of the MAC, demonstrating a broad heterogeneity of feline leprosy pathogens. Therefore, the term “feline leprosy” is becoming redundant and potentially inaccurate, both from a clinical point of view and because of the etiological agents involved. In fact, skin lesions caused by other NTM such as *M. avium* or MTBC members are often clinically indistinguishable from *M. lepraemurium* infections. Similarly to other strictly host-associated mycobacteria such as *M. leprae* and the closely related *M. lepromatosis*, they underwent reductive evolution [18]. Although some attempts at *M. lepraemurium* propagation in vitro have been successful, e.g., using 1% egg yolk medium, this can only be achieved under specific conditions due to its extremely fastidious nature [37,38,39]. Feline cutaneous mycobacteriosis is thought to be the result of infected bite or scratch wounds, surgical interventions or the lymphohematogenous spread of the pathogen [23]. In particular, *M. lepraemurium*, which causes leprosy in felines and rodents such as rats and mice, is most likely transmitted through bites during hunting activities. However, the possibility of an environmental niche, where the pathogen can persist without being in contact with infected individuals, has not yet been determined. This assumption appears to be compatible with the clinical and histopathological findings in wild rodents, where the pathogen is well adapted, and cats representing incidental hosts [10].

As for human leprosy, the lesions encountered in affected cats can be histologically differentiated into two different forms. Tuberculoid leprosy is frequently recognized as the benign form of the disease and is characterized by the presence of relatively small numbers of acid-fast bacilli (AFB), while lepromatous leprosy is associated with numerous, generally intracellular, AFB and is regarded as the malignant form [10,40]. The development of these different forms depends on various factors, including the genetic and immunological characteristics of the host, and on the virulence of the bacillus [40]. The numerous AFB observed in the fine-needle aspirates suggest a lepromatous type of the disease in the present case. Recent observations on confirmed and published cases showed that feline leprosy caused by *M. lepraemurium* typically affects young cats 1–3 years of age [10]. There is no known predisposition to NTM infections in Maine Coon cats. Single cases have recently been described [41]; however, these are anecdotal reports and there is not overrepresentation such as for Abyssinian and Siamese cats [42], or Bassett Hounds and Miniature Schnauzer dogs [43]. Cutaneous skin lesions are typically located on the head or forelimbs and, in contrast to *Candidatus* ‘M. tarwinense’, which has a tendency to induce ocular and periocular lesions, *M. lepraemurium* rarely affects the eyes [10,11]. The present clinical case shows, however, that small animal clinicians should add feline leprosy caused by *M. lepraemurium* to their differential diagnosis list when confronted with progressive scleral and periocular masses. Moreover, depending on the patient and owner’s compliance, the prognosis of *M. lepraemurium* infection might be good. 

To date, reports of feline leprosy caused by *M. lepraemurium* have been more common from temperate coastal areas than from regions with continental or alpine climates [8]. The detection of this pathogen in a cat living in the Swiss Alps, which show marked differences in climatic conditions compared to coastal areas, raises concerns about a possible extension of its geographical occurrence. The causes of this hypothetical extension may be numerous, including the mobility of infected cats, the adaptive capacity of the pathogen or climate changes. Nevertheless, it could be speculated that improved diagnostic capacities, enhanced disease awareness among clinicians and specialized laboratory setups have enabled the detection of a rare but endemic pathogen.

The combination of antimycobacterial agents effective against slowly growing NTM such as rifampicin, clofazimine or clarithromycin with new-generation fluoroquinolones, e.g., moxifloxacin or pradofloxacin, has been described as appropriate for treating the infection [44] There are no guidelines indicating antimicrobials of choice and the duration of therapy after surgical resection for feline leprosy. Common standard treatment includes a combination of two of the above-mentioned antimicrobials for at least 2 months after the disappearance of lesions, resulting in 3 to 6 months of therapy [8,10]. The spontaneous remission of a solitary lesion has been described as well [45]. It is unclear if the administration of a four-week course of rifampicin combined with clarithromycin had a therapeutic benefit or not. However, it is important to note that, even though, in the present report, not all the nodular lesions could be removed with wide surgical margins, the spreading of infection along tissue planes was not observed and the recurrence of skin and ocular disease was not detected six months after the cessation of antimicrobial therapy. The nodule present on the underside of the tongue showed spontaneous healing before antimicrobial treatment was initiated, suggesting that the immune defenses of the cat were functional and able to control the infection or, alternatively, that this specific lesion had another cause. 

Similarities between human, feline and murine leprosy have been reported, including in the clinical manifestation, parasitism of phagocytic cells and induction of a strong humoral immunity [46]. Moreover, several antigenic determinants shared by *M. leprae* and *M. lepraemurium* have been described [46]. To the author’s knowledge, naturally acquired *M. lepraemurium* infections have been observed exclusively in rats, mice and cats, but not in humans or any other mammalian species [47]. Hence, the zoonotic potential for pet owners and clinicians seems to be low. Nevertheless, it is recommended to wear gloves while handling cats with cutaneous lesions where mycobacterial disease is a differential diagnosis, especially if purulent discharge is present.

## 5. Conclusions

We report the successful treatment of a multifocal *M. lepraemurium* infection with ocular and cutaneous involvement in a cat. The initial suspicion of nodular skin disease of tumorous origin was excluded after the identification of acid-fast bacilli and detection of mycobacterial DNA using molecular methods. The infection was treated with a combination of surgical procedures and the administration of antimicrobial agents. The geographical origin of the cat, living in the Alpine area, suggests that the niche of *M. lepraemurium* is broader than previously thought.

## Figures and Tables

**Figure 1 pathogens-10-00687-f001:**
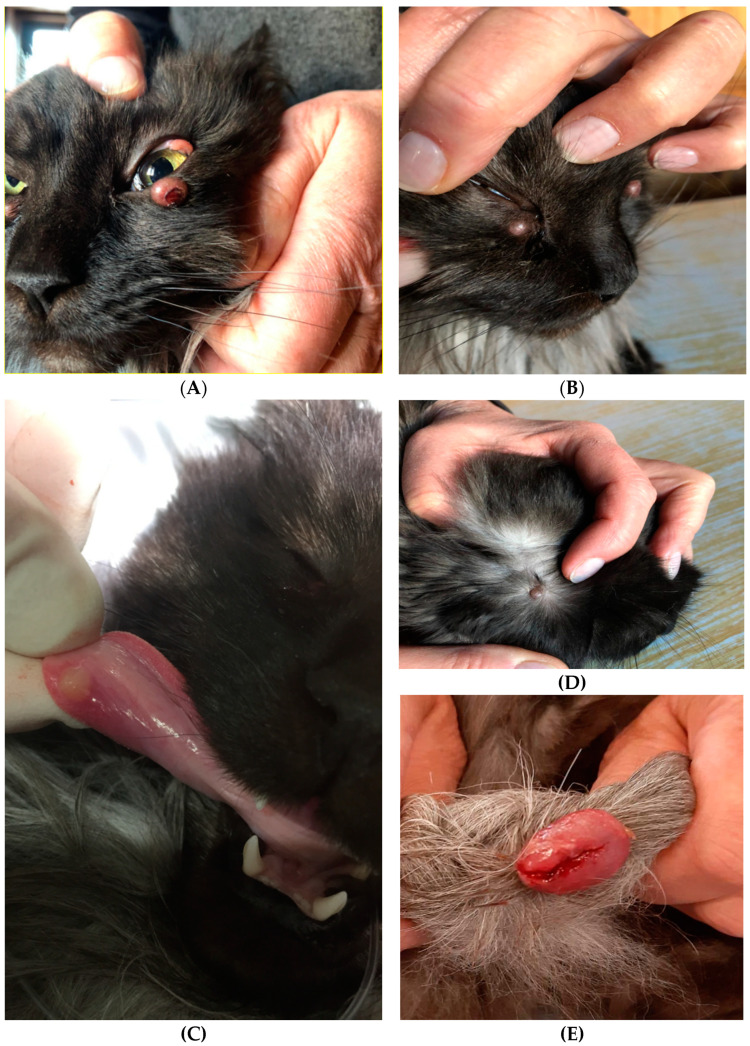
Left eye with ulcerative nodular skin lesion on the lower eyelid and non-ulcerative nodule on the dorsolateral sclera (**A**). Non-ulcerative nodule on the right medial canthus (**B**). Small non-ulcerative nodules on the ventral aspect of the tongue (**C**) and in the right ear-base region (**D**). Progressive ulcerative skin nodule in the distal part of the tail (**E**).

**Figure 2 pathogens-10-00687-f002:**
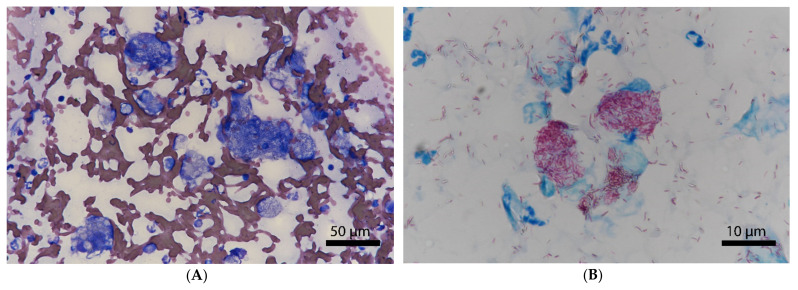
Cytological fine-needle aspirates of the lesions on the left lower eyelid and the sclera. Mixed inflammatory cell infiltrate consisting of neutrophils and of a large number of macrophages containing numerous intracytoplasmic rod-shaped, partly darkly blue-stained and partly unstained bacteria (**A**) that were acid-fast in the Ziehl–Neelsen stain (**B**).

**Figure 3 pathogens-10-00687-f003:**
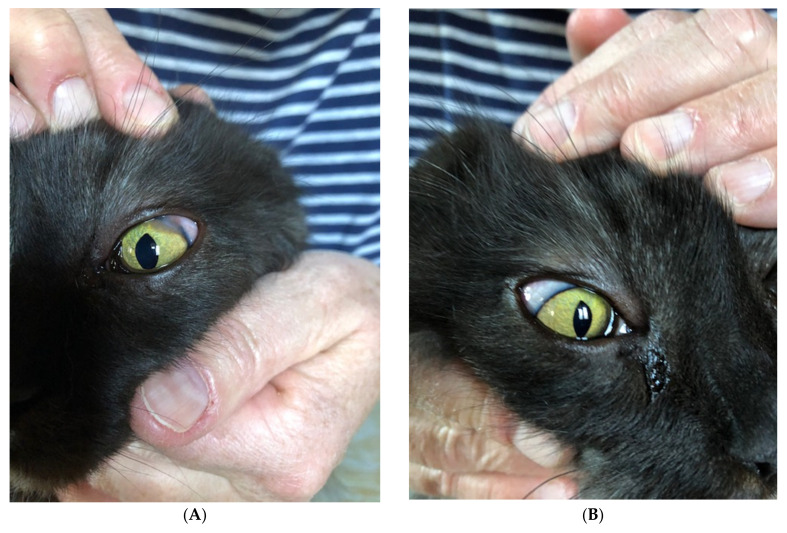
One year after surgical resection of the lesions and six months after discontinuation of the antimicrobials, the cat’s ocular lesions on the left (**A**) and right eye (**B**) were considered resolved.

## Data Availability

The data presented in this study are available on request from the corresponding author.

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
