# Peer review of "Unusual Presentation of Feline Leprosy Caused by Mycobacterium lepraemurium in the Alpine Region"

_pathogens, 2021, doi:10.3390/pathogens10060687_

Round 1

Reviewer 1 Report

This article present details on the diagnostic investigation and successful treatment of a case of feline M. lepraemurium infection in a Maine Coon cat in Switzerland. This case presents novel data, namely the demonstration of M. lepraemurium as causing ocular disease and should therefore be considered a differential diagnosis for cats presenting with such lesions, as well as the successful cultivation of this notoriously tricky to culture pathogen.

The article is well written, is easy to follow and is comprehensive in setting the context of and describing this case report. This reviewer only has minor comments that may hep to clarify some of the details of this case, and additional thoughts that could be incorporated into this report.

The term "feline leprosy", while one that is broadly accepted, is one this reviewer personally does not use due to the potential misunderstanding with human leprosy, where the peripheral nervous system is affected, whereas in cats this is not seen. Also, as addressed in the introduction, the term feline leprosy (which historically was reserved for cases where culture was unsuccessful) was attributed to presumed M. lepraemurium infection, however molecular methods have since identified many other species of non-tuberculous mycobacteria are responsible for "feline leprosy". Do the authors consider this term as becoming somewhat redundant, given the clinical similarities between cases of infection with "feline leprosy" pathogens, other NTM, as well as MTBC infections (this reviewer is aware of their recent publication of M. microti infections in cats in Switzerland)?

In the images, gloves were not worn by the individual holding the cat. The authors briefly touch upon the zoonotic potential of some mycobacterial infections. It may be prudent to include a comment in the discussion referring to the lack of gloves and that while in this case the inciting pathogen was non-zoonotic so there was minimal risk to human health, that the wearing of gloves would be recommended in cats with cutaneous lesions where mycobacterial disease is a differential diagnosis (especially if there is a purulent discharge). 

Although this is beyond the scope of this paper, this reviewer would like to ask whether the authors plan to conduct any further molecular characterisation of this isolate of M. lepraemurium, given the paucity of isolates that exist due to its extremely fastidious nature.

Some specific comments:

  • Line 31: consider adding "historically attributed to Mycobacterium lepraemurium infection" after "leprosy". This would help tie in to the further discussion in lines 35-36 and 43-46 about the wider breadth of this term and that it incorporates many other pathogens. If the authors wish to include this addition, the following "Mycobacterium tuberculosis" can then be shortened to "M. tuberculosis"
  • Line 60: a brief expansion of the zoonotic potential of some mycobacterial pathogens can be incorporated here, in particular those relevant to the cat. Cat-to-human transmission has been reported in cases of M. bovis infection (O'Connor et al., 2019), M. microti and some NTM are potentially zoonotic (including M. avium-complex, M. malmoense and M. kansasii), whereas M. lepraemurium is considered non-zoonotic
  • Lines 64-65: "first documented case of feline leprosy with ocular involvement in Europe". This statement should be reworded to something such as "the first documented case of M. lepraemurium infection with ocular involvement in Europe", as the introduction refers to a case from Italy (line 39) with ocular involvement, although the species of mycobacteria was not identified (ergo does this classify as feline leprosy?)
  • Lines 66 and 69: "Main" should be "Maine"
  • Line 69: add "with" between "presenting" and "nodular"
  • Line 73: remove "cat" from "The cat owner"
  • Line 85: the authors could consider adding the Köppen climate classification for the area this cat was from, as most previously reported cases (Australia, New Zealand, Great Britain) would have come from Cf climates
  • Line 86: add if the cat was a known hunter, or if there was known rodent access to the household
  • Line 93: does "complete blood counts" refer simply to haematology, or haematology and biochemistry? Was total and/or ionised calcium measured, and if so were these within reference intervals, increased or decreased?
  • Lines 87-98: it should be added whether this cat was receiving, or had recently been treated with any immunosuppressive or immunomodulatory medications, as these may increase the susceptibility to mycobacterial infection (as documented with humans)
  • Lines 110-111: include a comment to refer to the lack of sampling of the tongue, tail and ear base lesions for clarity that these were not investigated further
  • Lines 121-122: add whether treatment was undertaken for the tail and ear base lesions
  • Line 141: consider adding "approximately 120 days after surgery" between "molecular findings" and "the patient", to provide further clarity as to the timescale of surgery, culture and molecular identification, and specific antimicrobial therapy (I believe 120 days is correct based on the details provided in the text). Also, did the cat not receive any interim antimicrobial therapy while awaiting the results of culture?
  • Lines 143-144: this reviewer has a personal preference to report doses as mg/kg rather than the total dose, which would equate to 17.9mg/kg for rifampicin and 11.9mg/kg for clarithromycin
  • Line 144: "in vitro" should be italicised
  • Line 146: after "initial four weeks", consider adding a statement reporting whether any side effects were observed with both drugs, in particular with rifampicin. Some cats will tolerate higher doses such as this quite well, whereas others can have significant side effects at much lower doses
  • Line 151-152: I would move the statement regarding "the nodular lesion on the ventral aspect of the tongue which healed spontaneously" to earlier on in this paragraph. As it currently reads the cat received surgery and then a short course of antimicrobial treatment, and six months post-antibiosis the tongue lesion had 'spontaneously' resolved (which would not necessarily be spontaneous resolution if the cat had been on antibiotics). It is not until the discussion (lines 221-223) where it is clarified that the tongue lesion had resolved between the period of surgery and starting the cat on rifampicin and clarithromycin, hence resolution of this lesion was truly spontaneous. Also, what happened with the tail and ear base lesions? Did these resolve during the same period? Where they present when the cat was treated with antibiotics?
  • Lines 182-199: consider adding somewhere in this paragraph that cats with a diagnosis of M. lepraemurium typically have a good prognosis
  • Line 235: I would suggest using the word "multifocal" rather than "disseminated" to describe this infection. In this reviewer's opinion the term disseminated would infer spread to the thoracic and/or abdominal viscera, as well as lymph nodes, with histopathological changes of (pyo)granulomatous inflammation and the detection of mycobacteria (either through ZN-staining, culture or molecular identification)

Author Response

The term "feline leprosy", while one that is broadly accepted, is one this reviewer personally does not use due to the potential misunderstanding with human leprosy, where the peripheral nervous system is affected, whereas in cats this is not seen. Also, as addressed in the introduction, the term feline leprosy (which historically was reserved for cases where culture was unsuccessful) was attributed to presumed M. lepraemurium infection, however molecular methods have since identified many other species of non-tuberculous mycobacteria are responsible for "feline leprosy". Do the authors consider this term as becoming somewhat redundant, given the clinical similarities between cases of infection with "feline leprosy" pathogens, other NTM, as well as MTBC infections (this reviewer is aware of their recent publication of M. microti infections in cats in Switzerland)?

The authors agree with reviewer 1, the term "feline leprosy" is becoming redundant and somewhat inaccurate, both from a clinical point of view and because of the aetiological agents involved. In our experience, skin lesions caused by other NTM (such as M. avium) and MTBC members (especially M. microti) are often clinically indistinguishable from M. lepraemurium infections. Moreover, recent phylogenetic investigations demonstrated the heterogeneity of "feline leprosy" pathogens. Among veterinary practitioners however, the term "feline leprosy" is well known and allows a differentiation from other feline cutaneous mycobacteriosis. We have tried to put more emphasis on this issue in the Discussion section (Lines 181-186). Since this case report is also intended for practitioners and not just microbiologists, we decided to keep the term in this manuscript.

In the images, gloves were not worn by the individual holding the cat. The authors briefly touch upon the zoonotic potential of some mycobacterial infections. It may be prudent to include a comment in the discussion referring to the lack of gloves and that while in this case the inciting pathogen was non-zoonotic so there was minimal risk to human health, that the wearing of gloves would be recommended in cats with cutaneous lesions where mycobacterial disease is a differential diagnosis (especially if there is a purulent discharge). 

The authors are grateful for the extremely important remark concerning the lack of gloves in the images. This information has been added in the Discussion section (Lines 252-254).

Although this is beyond the scope of this paper, this reviewer would like to ask whether the authors plan to conduct any further molecular characterization of this isolate of M. lepraemurium, given the paucity of isolates that exist due to its extremely fastidious nature.

Yes, the authors would like to further characterize this isolate and we are seeking collaboration in the scientific community. Please do not hesitate to contact me ([email protected]) if you need any further information on the project or you would like to share your valuable experience.

Some specific comments:

  • Line 93: does "complete blood counts" refer simply to hematology, or hematology and biochemistry? Was total and/or ionized calcium measured, and if so were these within reference intervals, increased or decreased?

The cat had no haematological and biochemical significant alterations, unfortunately ionised calcium was not measured.

  • Line 141: consider adding "approximately 120 days after surgery" between "molecular findings" and "the patient", to provide further clarity as to the timescale of surgery, culture and molecular identification, and specific antimicrobial therapy (I believe 120 days is correct based on the details provided in the text). Also, did the cat not receive any interim antimicrobial therapy while awaiting the results of culture?

This is correct, the cat did not receive any antimicrobial therapy while awaiting the results of culture.

The authors are grateful to reviewer1 for the remaining important remarks and specific comments, these has been addressed in the manuscript.

Reviewer 2 Report

Dear authors,

the manuscript presents an interesting case of feline leprosy and is well-written. Nevertheless, I have some suggestions that may help to improve your paper.

Line 15: Change 'in the tail' to 'on the tail'.

Lines 66 and 69: Change 'Main Coon' to 'Maine Coon'.

Fig. 2: The inscriptions to the scale bars are far too small. In fact, they are virtually illegible. Morover, I wonder why the scale bar in Fig. 2B is not placed in the bottom right corner as in Fig. 2A.

It remains a little bit unclear what happened to the quite large lesion on the tail. Since it appears as a very exophytic mushroom-shaped mass, it sounds fairly implausible that it has fully disappeared without any surgical treatment.

It is very unfortunate that there have not been carried out any histopathological investigations, especially with regard to the mentioned two different types (tuberculoid vs. lepromatous) of leprosy-associated alterations and their immunological differences as well as taking the good clinical outcome into account. Histopathological findings would therefore really have been a significant benefit for the paper.

Author Response

It remains a little bit unclear what happened to the quite large lesion on the tail. Since it appears as a very exophytic mushroom-shaped mass, it sounds fairly implausible that it has fully disappeared without any surgical treatment.

The lesion was surgically removed before antimicrobial treatment was started. This information has been added at lines 158-159.

The authors are grateful to reviewer 2 for the remaining important remarks and specific comments, these has been addressed in the manuscript and Fig.2.

Reviewer 3 Report

A very interesting case report.  It's a rare infection. This is especially well documented. There are 2 novel findings - the location in the Swiss Alps, and the fact that they could culture the organism in liquid media. And the quality of case case management, microbiology is excellent. It does not add greatly to the literature, but the photos are terrific and it approximates the ideal way to treat such a case.

Author Response

The authors are grateful to the reviewer's valuable comments.

Reviewer 4 Report

A very interesting case report. 

My only comment relates to Lin 35/6 of the introduction - leprosy causing mycobacteria are by definition also non-tuberculous mycobacteria and so some authors e.g Malik R (ref 9) describe rapid or slow growing opportunistically pathogenic NTM and fastidious leprosy causing NTM as different classes rather than being "related to NTM". This is a minor distinction if the authors wanted to make the sentence clearer.

Author Response

The authors are grateful to reviewer 4 for the important remark. The mentioned sentence has been modified accordingly.